# Clinical and Peripheral Biomarkers in Female Patients Affected by Anorexia: Does the Neutrophil/Lymphocyte Ratio (NLR) Affect Severity?

**DOI:** 10.3390/nu15051133

**Published:** 2023-02-23

**Authors:** Alice Caldiroli, Davide La Tegola, Letizia Maria Affaticati, Francesca Manzo, Francesca Cella, Alberto Scalia, Enrico Capuzzi, Monica Nicastro, Fabrizia Colmegna, Massimiliano Buoli, Massimo Clerici, Antonios Dakanalis

**Affiliations:** 1Department of Mental Health and Addiction, Fondazione IRCCS San Gerardo dei Tintori, Via G.B. Pergolesi 33, 20900 Monza, Italy; 2Department of Medicine and Surgery, University of Milano Bicocca, Via Cadore 38, 20900 Monza, Italy; 3Department of Neurosciences and Mental Health, Fondazione IRCCS Ca’ Granda Ospedale Maggiore Policlinico, Via F. Sforza 35, 20122 Milan, Italy; 4Department of Pathophysiology and Transplantation, University of Milan, Via Festa del Perdono 7, 20122 Milan, Italy

**Keywords:** anorexia, severity of illness, inflammation, peripheral biomarkers, NLR, eating disorders

## Abstract

Anorexia Nervosa (AN) is a disabling disorder characterized by extreme weight loss and frequent chronicization, especially in its most severe forms. This condition is associated with a pro-inflammatory state; however, the role of immunity in symptom severity remains unclear. Total cholesterol, white blood cells, neutrophils, lymphocytes, platelets, iron, folate, vitamin D and B12 were dosed in 84 female AN outpatients. Mildly severe (Body Mass Index—BMI ≥ 17) versus severe (BMI < 17) patients were compared using one-way ANOVAs or χ^2^ tests. A binary logistic regression model was run to investigate the potential association between demographic/clinical variables or biochemical markers and the severity of AN. Patients with severe anorexia (compared to mild forms) were older (F = 5.33; *p* = 0.02), engaged in more frequent substance misuse (χ^2^ = 3.75; OR = 3.86; *p* = 0.05) and had a lower NLR (F = 4.12; *p* = 0.05). Only a lower NLR was predictive of severe manifestations of AN (OR = 0.007; *p* = 0.031). Overall, our study suggests that immune alterations may be predictive of AN severity. In more severe forms of AN, the response of the adaptive immunity is preserved, while the activation of the innate immunity may be reduced. Further studies with larger samples and a wider panel of biochemical markers are needed to confirm the present results.

## 1. Introduction

Eating disorders are behavioral conditions characterized by severe and persistent abnormal eating behaviors and associated distressing thoughts and emotions [1,2]. Anorexia nervosa (AN) is the eating disorder with the highest mortality rate of all psychiatric disorders due to medical complications associated with the illness, as well as suicide [3]. It is defined by an intense fear of weight gain and/or disturbed body image, determining severe dietary restriction and other weight-loss behaviors [4,5,6,7,8]. This debilitating and often chronic and relapsing condition has a prevalence of 0.8–6.3% in females and 0.1–0.3% in males [9]. AN is often associated with pronounced psychiatric comorbidity, emotional distress and functional impairment, as well as high rates of medical complications, especially in its most severe manifestations [10,11]. Medical sequelae of AN may involve all systems, including immunity. The gastrointestinal tract seems to be the most commonly affected system [12].

To date, it is widely recognized that nutritional deficiencies, malnutrition and starvation impair the human immune system, affecting cell-mediated immunity [13,14], as reflected by the depletion of leukocyte, lymphocyte and T-cell counts in restricting-type AN subjects without modifications of B cells [15,16]. Even though preliminary evidence demonstrated that AN was associated with immune system deficit, data regarding inflammatory alterations in patients affected by AN are still contrasting [17]. Notably, some authors pointed out that patients with AN, although severely malnourished, are relatively free from presenting an increased risk of common viral infections [15,18], suggesting that the adaptation process occurring in this disorder is multifaceted and worth studying.

Similar to other psychiatric disorders, AN appears to be characterized by immune system dysregulations. As concerns innate immunity, abnormalities in neutrophil chemotaxis, adherence and microbicidal activity have been suggested [19,20], as well as decreased levels of complement components [21,22] and Natural Killer (NK) cells [23,24]. Notably, these abnormalities are similar to those observed in primarily malnourished patients [25], unlike alterations found in the adaptive immune system. In particular, the CD4/CD8 ratio is seemingly increased in AN compared to primary malnutrition, likely due to a greater reduction in CD8 rather than CD4 cells counts [26,27]. On the other hand, fewer studies have focused on humoral immunity in AN, with generally inconclusive results [25].

Together with the complete blood count, a number of studies supported the clinical utility of measuring biochemical parameters that can be affected by AN-related malnutrition [28,29], such as vitamins (particularly vitamin D, vitamin B12 and folate), electrolytes, albumin and trace elements (e.g., copper, manganese, selenium and zinc) [30]. Among biochemical alterations, hypercholesterolemia was repeatedly observed [31] and a recent meta-analysis reported increased levels of total cholesterol, low-density lipoprotein (LDL), high-density lipoprotein (HDL) and triglycerides in acutely ill patients, with alterations in total cholesterol and LDL persisting after weight restoration [32].

Notably, a strong interplay exists between nutritional status and the immune–endocrine system [14]: chronic, low-grade inflammatory state has been already demonstrated in patients affected by AN [33]. For example, in the cytokine profile of anorexic patients, higher serum levels of Interleukin (IL)-1β, IL-6, Tumor Necrosis Factor (TNF)-α and IL-15 have been noted compared to healthy controls [34,35,36], as also reported by two meta-analyses [37,38]. These findings suggested the presence of a low-grade inflammatory state in AN, which is particularly evident in adult patients [25,38]. However, more recently, contradicting results have emerged regarding cytokine alterations, which seem to question the hypothesis of a uniform pro-inflammatory state across anorexic patients. Two recent papers [39,40] did not confirm the findings of increased TNF-α, IL-1β or IL-6 in AN. In addition, the study by Keeler and colleagues [41], demonstrated significantly lower TNF-α and IL-6 levels in affected individuals. Therefore, the amount and direction of cytokine changes in AN remain to be clarified.

It is also worth mentioning that alterations in gut microbiota have been described in AN, which appear not to normalize with weight gain [42,43], thus potentially representing more than an epiphenomenon. In addition, both clinical [44] and pre-clinical [45] research observed increased levels of oxidative stress in patients suffering from this disorder, contributing to the persistence of an over-inflammatory state in these individuals [46]. Finally, the neutrophil to lymphocyte ratio (NLR) and platelet to lymphocyte ratio (PLR), two well-established markers of low-grade inflammation, have been recently correlated with certain features of AN, such as a history of childhood maltreatment [47] and altered bone mineral density [48].

Even though it seems plausible that biochemical parameters may play a role in the onset and progression of AN [49], it is still unknown whether they represent state or trait markers of illness. Moreover, the correlation between peripheral biomarkers with illness severity has been poorly investigated until now, despite evidence of an association between AN severity and concentration of some inflammatory parameters [37].

In light of these considerations, the objective of the present study was to explore the relationship between a number of peripheral biomarkers with the severity of AN, as measured by body mass index (BMI).

## 2. Materials and Methods

### 2.1. Sample and Study Design

We conducted a cross-sectional study, recruiting outpatients consecutively admitted in a 20-month period (from January 2021 to August 2022) to the San Gerardo Hospital Outpatient Clinic for Eating Disorders.

Eligible subjects suffered from AN according to the Diagnostic and Statistical Manual of Mental Disorders (DSM) 5th edition (DSM-5) [4], were able and willing to provide informed consent. The study included participants aged between 17.5 and 45 years. Exclusion criteria were the following: (1) intellectual disability; (2) malnourishment due to severe organic disease; (3) pregnancy or breastfeeding; (4) taking vitamin B12 supplements; (5) patients in menopause. All participants provided written informed consent. The study was performed in compliance with the Helsinki Declaration of 1975, as revised in 2008, and the study protocol was approved by the ethics review board of San Gerardo Hospital. Diagnosis was determined via administration of the diagnostic items of the Italian version of the Eating Disorder Examination (EDE) Interview-17.0D [50].

We collected information about the following clinical variables: age, education, marital status, age at onset, duration of illness, duration of untreated illness (DUI), presence of substance use disorders, type of substance misuse, presence of psychiatric comorbidities, type of psychiatric comorbidities, presence of medical comorbidities, type of medical comorbidities, Body Mass index (BMI), presence of amenorrhea, current psychopharmacological treatment, pharmacological prescription at first visit to our psychiatric services and type of main pharmacological treatment.

DUI was considered as the time elapsed between the onset of the disorder and the first treatment with evidence of efficacy according to guidelines [51,52].

Blood samples were collected in fasting conditions in the morning during the first visit at our department to measure: total number of white blood cells, lymphocytes, neutrophils, platelets per μL, and plasma levels of vitamin D (ng/mL), vitamin B12 (pg/mL), total cholesterol (mg/dL), iron (mcg/dL) and folic acid (ng/mL). The NLR was then calculated dividing the total neutrophils count by the total lymphocytes count.

### 2.2. Statistical Analyses

Descriptive analyses of included variables were performed for the whole sample, and frequency with percentages and mean with standard deviation were calculated for qualitative and quantitative variables, respectively. The total sample was divided into two groups according to severity of AN: mild, when BMI ≥ 17 [4]; severe, when BMI < 17, the latter including moderately severe AN (16 < BMI < 16.99), severe AN (15 < BMI < 15.99) and extremely severe AN (BMI < 14.99) according to the DSM-5 severity criteria [4]. It was not possible to compare the 4 groups identified by DSM criteria in light of the small sample size.

The two groups were compared for quantitative variables (including biochemical values) and for qualitative ones using, respectively, univariate analyses of variance (one-way ANOVAs) and χ^2^ tests. Then, a logistic regression model was performed, including the significant variables from univariate analyses as independent variables and presence of moderate–severe (versus mild) AN as a dependent one. The goodness of the model was assessed by Omnibus and Hosmer–Lemeshow tests. The level of statistical significance was set at *p* ≤ 0.05.

The aforementioned statistical analyses were performed through The Statistical Package for Social Sciences (SPSS) for Windows (version 28.0, Milan, Italy).

## 3. Results

The whole sample consisted of 84 female patients. Forty-seven (56.0%) patients were classified as suffering from a mild form of AN, 37 (44.0%) with a moderate–severe manifestation of the illness. The mean age was 23.31 (±7.38) years. Table 1 summarizes the clinical and biochemical data, and differences between the two groups.

Patients affected by severe AN, compared to their counterparts, were older (*p* = 0.024) and had a lower NLR (*p* = 0.048). Moreover, subjects with severe AN (compared to the others): showed a trend in having a longer duration of illness (*p* = 0.088) and DUI (*p* = 0.085), higher vitamin B12 plasma levels (*p* = 0.078) and more frequent substance use disorders (*p* = 0.053).

The goodness-of-fit test (Hosmer and Lemeshow Test: χ^2^ = 5.328; df = 8; *p* = 0.722) showed that the model including age, duration of illness, DUI, substance misuse, NLR and vitamin B12 plasma levels as possible predictors of AN severity was reliable, allowing for a correct classification of 81.3% of the cases. In addition, the model was significant overall (Omnibus test: χ^2^ = 18.126; df = 6; *p* = 0.006).

Higher NLR plasma levels were associated with mild rather than severe AN (*p* = 0.031) (Table 2).

## 4. Discussion

The results of our study suggest that the severity of AN might be influenced by alterations in different biological systems. In particular, the main finding of our study was that a greater severity of AN is associated with a lower NLR.

NLR is an easily obtainable parameter and a documented marker of physiologic stress and low-grade systemic inflammation [53]. NLR rises during stressful situation when a shift from adaptive to innate immunity is usually observed [54]. The normal range of NLR in adult, non-geriatric, healthy populations varies between 0.78 and 3.53 (mean 1.65 ± 1.96 SD [55] or 1.70 ± 0.70 [56]. In our sample, the mild-severity group of anorexic patients had a mean value of 1.74 ± 0.94 (range 0.40–5.42), in line with reports in the general population. On the other hand, severe anorexic patients presented with a mean NLR plasma level of 1.23 ± 0.66 (range 0.35–3.21), and hence with a slight shift to lower values. Although neither group showed a substantially different NLR compared to that reported in the general population, our results did suggest significant variations as a function of AN severity. The reason for a significantly lower NLR in more severe AN patients is yet to be elucidated.

One hypothesis is based on the generally accepted evidence that starvation and malnutrition, as well as single nutrient deficiencies, may alter the production of leukocytes from the bone marrow. As reported by several authors, anemia, leukopenia and thrombocytopenia are frequent complications of anorexia nervosa [14,16] and may be a consequence of the degeneration of the bone marrow. Degenerative processes include serous fat atrophy and gelatinous transformation, which appear to be related to the lack of carbohydrates in the diet of AN patients [57]. The particular kind of malnutrition of patients with AN differs from the protein-energy malnutrition (PEM), which represents the most common cause of human immunodeficiency [58]; in fact, differently from PEM, in AN, the risk of common viral infections is not increased [15]; CD4+ T-cell counts are usually preserved with reduced percentages of CD8+ T-cells [26,27,33]. The observation that nutrition rehabilitation ameliorates leukocyte alterations supports the hypothesis that hematological alterations in AN are driven by malnourishment [23]. It is, however, worth mentioning that NLR seems to increase, rather than decrease, in other forms of malnutrition. Indeed, NLR was found to be significantly higher in geriatric outpatients who were malnourished or at risk of malnourishment [59]. Similarly, an NLR ≥ 5.0 was found to be predictive of nutritional risk in cancer inpatients [60], whilst an NLR ≥ 2.62 identified protein-energy wasting in individuals with a diagnosis of chronic kidney disease [61].

Our results appear to be consistent with the available literature, although the data are still poor and controversial. Lambert and collaborators [62] found a correlation between body mass fat and leukocyte count, albeit this finding was not confirmed by other authors [63]. In addition, a greater severity of weight loss was reported to correlate with the degree of bone marrow failure [64] and with a higher CD4/CD8 ratio [65], suggesting that, with the progression of weight loss, lymphocyte production is prioritized over other immune cells in order to preserve the adaptive immune system functioning. More recently, Saito and collaborators [66] demonstrated that, while the CD8 T-cell count did not vary according to the severity of nutritional status, the absolute lymphocyte count and the CD4 proportion presented a positive and a negative correlation with BMI, respectively.

Considering the interplay between cytokines, neuropeptides and neurotransmitters, a second possible explanation of a lower NLR in severe AN patients compared to their counterparts is based on the assumption that inflammation plays a specific role in the etiology of AN. This hypothesis has been already suggested [25,33,67], although, to date, the role of inflammatory markers in AN is still unclear, as is the state or trait nature of this alteration. Several studies demonstrated that, despite the low BMI, blood levels of different pro-inflammatory cytokines such as TNF-α, IL-1α and IL-6 were significantly increased in AN compared to HCs [37,38,68]. Nevertheless, few data have been published until now regarding the role of illness severity in the pro-inflammatory state of patients affected by AN [69]. Consistently with our findings, Nilsson and colleagues [40] found a positive association between BMI and IL-6 plasma levels in AN, while other authors reported that chronic AN results were associated with a non-inflammatory status [70]. Globally, these preliminary findings support the hypothesis of a prominent pro-inflammatory status at the beginning of the disorder (mild AN) and of the contribution of immune dysregulation in the persistence of symptoms of AN, similar to what happens in other psychiatric conditions [71]. It is plausible that the progression of AN favors the shift from an over-activation of innate immunity to a predominance of adaptive immunity concomitantly with a change in the cell composition of bone marrow [72].

A trend of substance use disorders seems to characterize severe versus mild forms of AN. Our finding is consistent with previous literature reporting that the risk of substance use disorders in AN is influenced by subtype (i.e., it is more prevalent in the binge-eating/purge type than in restrictive type) [73] and by severity of symptoms, particularly in adolescents and for alcohol misuse [74]. The comorbidity of substance use disorders in subjects affected by AN is associated with a worse prognosis and an increased risk of somatic diseases and mortality [75,76].

Finally, vitamin B12 plasma levels resulted as higher in patients with a severe versus a mild form of AN, with a trend to statistical significance. Different authors reported that vitamin B12 plasma levels were normal or even elevated in subjects affected by AN [30,77]. Similar to our findings, Corbetta and colleagues [78] demonstrated that the levels of vitamin B12 and folate depended on the severity of illness, leading to the hypothesis that these vitamins may represent early markers of liver dysfunction. Another recent study supported this statement, demonstrating changes in vitamin B12 plasma levels concomitantly with liver dysfunction in acute AN patients [79].

These results should be interpreted in light of several study limitations. First, the sample size was relatively small. Second, the recruited subjects were all outpatients and this could represent a recruitment bias, as extremely severe AN patients were largely excluded. Third, peripheral markers were chosen a priori, being adherent to the routinely assessed parameters, but at the same time excluding, for example, other vitamins or cytokine plasma levels. In this regard, we are planning to expand the present study measuring circulating levels of TNF-α, IL-6 and CRP, as well as other inflammatory cytokines and antioxidant factors. In fact, it has been demonstrated that there is an antioxidant deficiency in AN [44,80]. Although modifications of the lipoprotein profile, the structure of phospholipids and constituents of myelin seemed to be related to specific clinical features of AN, such as body image distortion [81,82], the neurobiological underpinnings of AN are still uncertain and need to be further explored. Fourth, we measured AN severity using BMI as single parameter, according to the DSM-5; however, severity of AN is a more complex concept which includes, for example, rapidity of weight loss and body composition. Finally, some supplements as well as substance misuse might have influenced the values of some biochemical markers; nevertheless, we excluded patients taking supplements of vitamin B12.

## 5. Conclusions

Our study demonstrated that women with AN presented different immune and biochemical alterations according to the degree of illness severity. In particular, NLR was lower in severe AN patients than in the mildly severe group. This finding supports the hypothesis of a dysregulated immune system in AN, although it remains difficult to determine whether these immunological changes are directly involved in the development and maintenance of the disorder or secondary to malnutrition.

Further studies on larger samples and including a wider range of inflammatory and nutritional markers (e.g., cytokines and other vitamins) are needed to better elucidate the underlying biological dysfunctions associated with the onset of AN. The identification of potential biomarkers of AN severity may help to better characterize the different phases of the disorder, contributing to improve preventive and treatment strategies in the era of personalized medicine.

## Figures and Tables

**Table 1 nutrients-15-01133-t001:** Clinical variables and peripheral biomarkers of the total sample and of the two groups identified according to the severity of anorexia.

Variables		Mild Severity N = 47	Moderate-Extreme Severity N = 37	Total Sample N = 84	F or χ^2^	*p* Value
Age		21.70 (±5.04)	25.35 (±9.24)	23.31 (±7.38)	5.33	**0.024**
Age at onset		16.96 (±2.80)	18.78 (±8.32)	17.78 (±5.99)	1.913	0.170
Duration of illness (months)		46.11 (±55.36)	71.06 (±72.85)	56.85 (±64.26)	2.994	0.088
DUI (months)		24.80 (±30.05)	41.63 (±55.02)	32.16 (±43.32)	3.048	0.085
Amenorrhea		11 (23.4%)	14 (37.8%)	25 (29.8%)	2.063	0.151
Presence of substance use disorders		3 (6.5%)	7 (21.2%)	10 (12.7%)	3.751	**0.053 ^1^**
Type of misuse	Alcohol	1 (2.2%)	5 (15.2%)	6 (7.7%)	5.472	0.065
	Cannabis	1 (2.2%)	2 (6.1%)	3 (3.8%)		
Presence of psychiatric comorbidity		18 (38.3%)	16 (43.2%)	34 (40.5%)	0.210	0.647
Type of psychiatric comorbidity	Unipolar depression	8 (17%)	8 (21.6%)	16 (19%)	4.148	0.528
	Anxiety disorders	6 (12.8%)	2 (5.4%)	8 (9.5%)		
	Personality disorders	3 (6.4%)	3 (8.1%)	6 (7.1%)		
	OCD	1 (2.1%)	1 (2.7%)	2 (2.4%)		
	Others	0 (0.0%)	2 (5.4%)	2 (2.4%)		
Presence of medical comorbidity		12 (26.1%)	8 (22.9%)	20 (24.7%)	0.112	0.738
Type of medical comorbidity	Chronic kidney disease	1 (2.1%)	0 (0.0%)	1 (1.2%)	8.905	0.711
	Hypertension	1 (2.1%)	0 (0.0%)	1 (1.2%)		
	Gastroesophageal reflux disease	1 (2.1%)	1 (2.9%)	2 (2.4%)		
	Hypothyroidism	1 (2.1%)	0 (0.0%)	1 (1.2%)		
	Hyperthyroidism	0	1 (2.9%)	1 (1.2%)		
	Cancer	1 (2.1%)	0 (0.0%)	1 (1.2%)		
	Anemia	2 (4.3%)	1 (2.9%)	3 (3.7%)		
	Other endocrine disorders	2 (4.3%)	1 (2.9%)	3 (3.7%)		
	Other hematological disorders	1 (2.1%)	1 (2.9%)	2 (2.4%)		
	Other gastrointestinal disorders	2 (4.3%)	0 (0.0%)	2 (2.4%)		
	Other neurological disorders	1 (2.1%)	1 (2.9%)	2 (2.4%)		
	Others	0 (0.0%)	2 (5.7%)	2 (2.4%)		
Marital status	Single	31 (72.1%)	18 (69.2%)	49 (71%)	0.064	0.800
	In a relationship	12 (27.9%)	8 (30.8%)	20 (29%)		
Education	Secondary school	15 (32.6%)	8 (24.2%)	23 (29.1%)	1.832	0.400
	High school	27 (58.7%)	19 (57.6%)	56 (58.2%)		
	University degree	4 (8.7%)	6 (18.2%)	10 (12.7%)		
N. of siblings	0	5 (10.6%)	8 (21.6%)	13 (15.5%)	4.085	0.130
	1	20 (42.6%)	19 (51.4%)	39 (46.4%)		
	>1	22 (46.8%)	10 (27.0%)	32 (38.1%)		
Presence of family history of psychiatric disorders	Yes	16 (35.6%)	15 (42.9%)	31 (38.7%)	0.442	0.506
	No	29 (64.4%)	20 (57.1%)	49 (61.3%)		
Type of family history of psychiatric disorders	Unipolar depression	6 (13.3%)	4 (11.4%)	10 (12.5%)	9.755	0.283
	Anxiety disorders	4 (8.9%)	1 (2.9%)	5 (6.3%)		
	Schizophrenia spectrum disorders	1 (2.2%)	0 (0.0%)	1 (1.3%)		
	Dissociative disorder	1 (2.2%)	0 (0.0%)	1 (1.3%)		
	Personality disorder	0 (0.0%)	1 (2.9%)	1 (1.3%)		
	Eating disorders	2 (4.4%)	6 (17.1%)	8 (10%)		
	OCD	1 (2.2%)	0 (0.0%)	1 (1.3%)		
	Others	1 (2.2%)	3 (8.6%)	4 (5%)		
Current pharmacological treatment	Yes	13 (27.7%)	13 (35.1%)	26 (31%)	0.541	0.462
	No	34 (72.3%)	24 (64.9%)	58 (69%)		
Pharmacological prescription at first presentation to psychiatric services	Yes	18 (38.3%)	12 (32.4%)	30 (35.7%)	0.310	0.578
	No	29 (61.7%)	25 (67.6%)	54 (64.3%)		
Type of prescription	Antidepressant	15 (31.9%)	9 (24.3%)	24 (28.6%)	1.832	0.608
	Second-generation antipsychotic	2 (4.3%)	3 (8.1%)	5 (6%)		
	GABAergic drug	1 (2.1%)	0 (0.0%)	1 (1.2%)		
BMI		18.44 (±1.30)	15.11 (±1.35)	16.97(±2.12)	130.38	**<0.001**
Vitamin D (ng/mL)		25.58 (±9.48)	23.15 (±6.88)	24.69 (±8.58)	0.550	0.464
Total cholesterol (mg/dL)		162.33 (±23.61)	178.44 (±30.93)	167.17 (±26.55)	2.435	0.130
Vitamin B12 (pg/mL)		437.00 (±129.80)	573.38 (±326.91)	484.92 (225.27)	3.287	0.078
Iron (mcg/dL)		97.0538 (±49.21)	87.58 (±36.93)	94.06 (±45.40)	0.351	0.557
Folate (ng/mL)		9.05 (±7.63)	7.96 (±3.86)	8.71 (±6.63)	0.219	0.642
White blood cells (per µL)		6045.15 (±1612.94)	5303.16 (±2231.00)	5774.04 (±1875.96)	1.920	0.172
Platelets (per µL)		230727.27 (±48815.51)	232842.10 (±46746.67)	231500 (±47618.38)	0.023	0.879
Lymphocytes (per µL)		2095.94 (±651.75)	2221.11 (±827.36)	2141 (±100.98)	0.349	0.557
Neutrophils (per µL)		3322.19 (±1331.65)	2612.22 (±1600.53)	3066.6 (±206.36)	2.829	0.099
NLR		1.74 (±0.94)	1.23 (±0.66)	1.56 (±0.88)	4.124	**0.048**

Legend: BMI—body mass index; DUI—duration of untreated illness; NLR—neutrophil-lymphocyte ratio; OCD—obsessive compulsive disorder. Mean (±standard deviation) for continuous variables; frequencies (percentage) for qualitative variables. In **bold** statistically significant differences. ^1^ borderline statistical significance.

**Table 2 nutrients-15-01133-t002:** Summary of the results from binary logistic regression model.

Variables	B	S.E.	*p* Value	OR	95% C.I.	
					Inferior	Superior
Age	0.014	0.383	0.970	1.014	0.479	2.148
Duration of illness	0.024	0.036	0.513	1.024	0.954	1.099
DUI	0.015	0.043	0.717	1.016	0.934	1.104
Substance misuse	2.681	1.974	0.175	3.859	0.917	16.247
NLR	−5.009	2.325	**0.031**	0.007	0.000	0.637
Vitamin B12	0.003	0.003	0.333	1.003	0.997	1.009

Legend: B = regression coefficient; C.I. = confidence interval; DUI = duration of untreated illness; NLR = neutrophil to lymphocyte ratio; OR = odds ratio; S.E. = standard error. The dependent variable is represented by the presence of more severe anorexia (moderate–extreme severity). In **bold** statistically significant differences (*p* < 0.05).

## Data Availability

Data sharing is not applicable to this article.

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
