# Peer review of "Clinical and Peripheral Biomarkers in Female Patients Affected by Anorexia: Does the Neutrophil/Lymphocyte Ratio (NLR) Affect Severity?"

_nutrients, 2023, doi:10.3390/nu15051133_

Round 1

Reviewer 1 Report

The present work shows that women with AN have some biochemical and immune alterations associated with the degree of , without a doubt it is an interesting and important work for the area of ​​study.I believe that with a few small changes the work would be even more interesting. I believe that some measurement of the inflammatory state of the patients, such as circulating levels of TNF-alpha, IL 6 and C-reactive protein, would greatly improve the discussion.Another suggestion would be to measure levels of oxidative damage to circulating proteins and lipids, as well as levels of oxidized lipoproteins,or even a dosage of the total antioxidant potential of the plasma
Table legends should be more explanatory

Author Response

Thank you for your observation and suggestions. We discussed the importance of studying antioxidant markers and more inflammatory parameters. We added in the limits these aspects and we reported our plan to expand this study in order to measure also other inflammatory cytokines and antioxidants.

Reviewer 2 Report

This paper shows that neutrophil/lymphocyte ratio (NLR) was lower in patients with severe Anorexia Nervosa (AN) in 84 female patients. It is interesting that the severity of AN can be evaluated and predicted by leucocyte types without employing serum proteins and nutrients. There are a few minor issues to be addressed.

1) Please discuss which NLR was abnormal about severe and milder AN patients compared to that in normal status.

2) Table 1: It is unclear whether menopause was considered in the counting of patients with amenorrhea. Aged patients seem to be participating in this study (as indicated in line 90).

3) It is better to discuss whether primary malnutrition involves changes in NLR.

Author Response

1) Please discuss which NLR was abnormal about severe and milder AN patients compared to that in normal status.

Thank you for your suggestion. We added a paragraph regarding normal values of NLR in the Discussion.

2) Table 1: It is unclear whether menopause was considered in the counting of patients with amenorrhea. Aged patients seem to be participating in this study (as indicated in line 90).

Thank you for your notice. We specified in the Inclusion criteria that we excluded women in menopause. We checked the dataset and the oldest patient was 45 years old. We modified the inclusion criterion regarding age for recruitment to make it clearer.

3) It is better to discuss whether primary malnutrition involves changes in NLR.

Thank you for your observation. We discussed NLR changes in primary malnutrition as you suggested.